# Overexpression of Two Members of D7 Salivary Genes Family is Associated with Pyrethroid Resistance in the Malaria Vector *Anopheles Funestus* s.s. but Not in *Anopheles Gambiae* in Cameroon

**DOI:** 10.3390/genes10030211

**Published:** 2019-03-12

**Authors:** Emmanuel Elanga-Ndille, Lynda Nouage, Achille Binyang, Tatiane Assatse, Billy Tene-Fossog, Magellan Tchouakui, Daniel Nguete Nguiffo, Helen Irving, Cyrille Ndo, Parfait Awono-Ambene, Charles S. Wondji

**Affiliations:** 1Centre for Research in Infectious Diseases (CRID), P.O. BOX 13591, Yaoundé, Cameroon; lynda.djounkwa@crid-cam.net (L.N.); achille.binyang@crid-cam.net (A.B.); tatianeassatse@gmail.com (T.A.); billy.tene@crid-cam.net (B.T.-F.); magellan.tchouakui@crid-cam.net (M.T.); daniel.nguiffo@crid-cam.net (D.N.N.); cyrille.ndo@crid-cam.net (C.N.); 2Vector Group, Liverpool School of Tropical Medicine, Pembroke Place, Liverpool L3 5QA, UK; helen.irving@lstmed.ac.uk; 3Department of Animal Biology and Physiology, Faculty of Science, University of Yaoundé 1, P.O. Box 812, Yaoundé, Cameroon; 4Department of Animal Biology, Faculty of Science, University of Dschang, P.O. Box 67, Dschang, Cameroon; 5Malaria Research Laboratory, Organisation de Coordination pour la lutte Contre les Endémies en Afrique Centrale (OCEAC), P.O. Box 288, Yaoundé, Cameroon; hpaawono@yahoo.fr; 6Department of Biological Sciences, Faculty of Medicine and Pharmaceutical Sciences, University of Douala, P.O. Box 24157, Douala, Cameroon

**Keywords:** D7 salivary proteins, gene expression, insecticide resistance, pyrethroids, *Anopheles gambiae*, *Anopheles funestus*, malaria disease

## Abstract

*D7* family proteins are among the most expressed salivary proteins in mosquitoes. They facilitate blood meal intake of the mosquito by scavenging host amines that induce vasoconstriction, platelet aggregation and pain. Despite this important role, little information is available on the impact of insecticide resistance on the regulation of *D7* proteins and consequently on the blood feeding success. In this study, real-time quantitative polymerase chain reaction (qPCR) analyses were performed to investigate how pyrethroid resistance could influence the expression of genes encoding *D7* family proteins in *Anopheles gambiae* and *Anopheles funestus s.s.* mosquitoes from Elon in the Central Cameroon. Out of 328 collected mosquitoes, 256 were identified as *An. funestus sl* and 64 as *An. gambiae sl*. Within the *An. funestus* group, *An. funestus s.s*. was the most abundant species (95.95%) with *An. rivulorum, An. parensis* and *An. rivulorum*-like also detected. All *An. gambiae s.l* mosquitoes were identified as *An. gambiae.* High levels of pyrethroid resistance were observed in both *An. gambiae* and *An. funestus* mosquitoes. RT-qPCR analyses revealed a significant overexpression of two genes encoding *D7* proteins, *D7r3* and *D7r4*, in pyrethroids resistant *An. funestus*. However, no association was observed between the polymorphism of these genes and their overexpression. In contrast, overall *D7* salivary genes were under-expressed in pyrethroid resistant *An. gambiae*. This study provides preliminary evidences that pyrethroid resistance could influence blood meal intake through over-expression of *D7* proteins although future studies will help establishing potential impact on vectorial capacity.

## 1. Introduction

In the early 2000s, thanks to the Abuja Declaration and the achievement of the Millennium Development Goals, immense efforts have been made to control malaria, with the goal to ultimately eliminate its transmission worldwide. These efforts have led to a significant and substantial progress in reducing malaria morbidity and mortality, despite the slight upward trend observed between the years 2016 and 2017 [1]. This remarkable reduction of malaria burden is strongly associated with an increased coverage of people at risk with insecticide-treated bed nets (ITNs), long-lasting insecticidal nets (LLINs) and to a lesser extent indoor residual spraying (IRS) with insecticides [2]. Unfortunately, this massive use of pyrethroids is also associated with the rapid expansion of insecticide resistance in both *Anopheles funestus* and *Anopheles gambiae s.l.*, the two major African malaria vectors. To maximize the effectiveness of current and future insecticide-based interventions, it is imperative to better understand the impact of insecticide resistance on mosquito populations including their vectorial capacity.

Little is currently known about the impact of resistance on the ability of malaria vectors to take a blood meal which is a key parameter of vectorial capacity and malaria transmission. Blood meal intake has been shown to be influenced by specific salivary proteins including the *D7* proteins family [3,4]. Indeed, the members of the *D7* salivary family are among the most abundantly expressed proteins in the salivary gland of blood-sucking Diptera [3,4]. Through their vasodilation, anticoagulant and anti-platelet aggregation properties, these proteins are essential for the success of the mosquito blood meal intake [3,4].

Interestingly, several studies assessing the transcriptomic changes associated with insecticide resistance have detected beside the overexpression of detoxification genes, a differential expression of some genes encoding for salivary proteins such as, *D7* salivary proteins [5,6,7,8,9,10].

Despite the potentially important role of *D7* in blood meal intake, far less information is available about the impact of pyrethroid resistance on the expression of the D7 family proteins and others mosquito salivary gland bioactive molecules implicated in the blood meal intake and in *Plasmodium* transmission. Indeed, the little information available about how resistance to pyrethroid impacts the expression of *D7* family salivary proteins are provided by results of microarray analysis of the transcriptome of resistant *Anopheles funestus* and *Anopheles gambiae* sl mosquitoes [5,6,7,8,9,10]. The *D7* salivary proteins family in *Anopheles* mosquito species was described to be composed with multiple genes. These genes can be distinguished in short and long forms [4,11,12,13]. The *An. gambiae* genome carries eight members of the *D7* gene family, among which, three genes encode long *D7* proteins (*D7L1*, *D7L2* and *D7L3*) and five code for short D7 proteins (*D7r1*, *D7r2*, *D7r3*, *D7r4* and D7r5) [3,12,14]. In *An. funestus*, members of D7 gene family are currently known to be constituted by five genes coding for short proteins (*D7r1*, *D7r2*, *D7r3*, *D7r4* and AFUN016458, an orthologous of *An. gambiae D7r5* gene) and two coding for long proteins (*D7L1* and *D7L2*) [4,11,12,13]. Thus, the expression pattern of members of this family in relation to insecticide remains to be established as well as the associated polymorphisms.

The present study tends to fill this knowledge gap by reporting the results of differential gene expression of several members of the *D7* salivary gene family and genetic diversity in relation to pyrethroid resistance in both *Anopheles funestus* and *Anopheles gambiae* wild mosquitoes from an area with high coverage of LLINs in Cameroon.

## 2. Methods

### 2.1. Study Area and Mosquito Collection

*An. funestus* and *An. gambiae* mosquitoes were collected in the locality of Elon (N4.23051° E11.60120°), a forest area in the Central Cameroon. This rural area situated about 50 km from Yaoundé falls within a Guinea-type climate with high humidity and precipitation and rainfall averaging 1000–2000 mm each year. The village is populated by about 300 inhabitants who are mainly subsistence farmers cultivating maize, cassava, yams, groundnut and vegetables. In this locality, there is also a stone quarry belonging to a company specializing in civil engineering works. Activities of this quarry have led to the transformation of a small stream into an artificial lake potentially suitable for the development of some *Anopheles* mosquito species, particularly those of *An. funestus* group. Collections were implemented during the rainy season in June 2018. Following the acquirement of a verbal consent from the chief of the village and the household owners, indoor resting adult female *Anopheles* mosquitoes were collected on the walls and on the roof of different houses across the village between 6:00 and 10:00 AM during 3 consecutive days using electric aspirators (Rule In-Line Blowers, Model 240). Caught mosquitoes were kept in paper cups in a cool place prior to transport to the insectary at Centre for Research in Infectious Diseases (CRID) in Yaoundé. After species identification based on morphological keys [15], blood-fed female mosquitoes were kept in cages for four days until eggs became mature. Gravid mosquitoes were allowed to oviposit according to the forced egg-laying method as previously described [16]. Thereafter, F1 adults were randomly mixed in cages for subsequent experiments. All female mosquitoes (oviposited, non-oviposited and unfed) were conserved in microtubes containing desiccant for further experiments such as PCR identification, genotyping of some resistance markers and determination of the *Plasmodium* sporozoite infection rate of both *An. funestus* and *An. gambiae* mosquitoes.

### 2.2. Molecular Species Identification

Genomic DNA was extracted from collected *An. funestus* and *An. gambiae* mosquitoes using the Livak protocol [17]. The protocol previously described by Koekomoer et al. [18] was used for the identification of the members of *An. funestus* group, whereas a SINE PCR [19] was performed for those belonging to *An. gambiae* complex.

### 2.3. Plasmodium sporozoite Infection Rate

*Plasmodium* infection rate of both *An. funestus* group and *An. gambiae* complex mosquitoes was determined using a TaqMan assay to detect four *Plasmodium* species (*Plasmodium falciparum*, *Plasmodium vivax*, *Plasmodium ovale* and *Plasmodium malariae*) [20] from genomic DNA extracted from individual field-collected mosquitoes.

### 2.4. Insecticide Susceptibility Assays

World Health Organization (WHO) insecticide susceptibility test-kits and standard procedures [21] were used to assess the susceptibility to insecticides. For *Anopheles funestus*, F1 female mosquitoes aged 3–5 days were exposed for one hour to: three pyrethroids including permethrin (0.75%), deltamethrin (0.05%) and etofenprox (0.05%); two carbamates: bendiocarb (0.1%) and propoxur (0.1%); one organochlorine: dieldrin (4%); and two organophosphates: malathion (5%) and fenitrothion (1%). For *Anopheles gambiae*, because of low mosquito densities, only permethrin (0.75%) and deltamethrin (0.05%) were tested. For each insecticide, assays were carried out with five replicates of 20–25 mosquitoes: four batches were exposed to insecticide-impregnated papers and one was exposed to untreated filter paper as a control. The mortality rate was determined 24 h after exposure and the resistance status was evaluated according to the WHO criteria [21]. Dead mosquitoes were kept in silica gel whereas alive mosquitoes were stored in 1.5 mL Eppendorf tubes containing RNA-later and kept in −80 °C.

Piperonyl butoxide (PBO) synergist assays were also performed to assess the contribution of cytochrome P450s in the resistance to deltamethrin and permethrin in *An. funestus* and *An. gambiae* mosquitoes from Elon. After been pre-exposed to 4% PBO for one hour, F1 adult female mosquitoes (3–5 days old) were immediately exposed to permethrin (0.75%) and to deltamethrin (0.05%) for 1 h. Final mortality was recorded after 24 h and compared to the results obtained using deltamethrin and permethrin without PBO.

Additionally, cone bioassays were carried out following the WHO guidelines [22] in order to assess the bio-efficacy of common bed nets recommended by WHO against *An. funestus* and *An. gambiae* mosquito populations from Elon. Five replicates of ten unfed *An. gambiae* and *An. funestus* F_1_ females (3–5 days old) were exposed for 3 min to netting pieces cut from 4 commercial nets: PermaNet^®^ 2.0 (deltamethrin 1.8 g/kg), (Vestergaard, Lausanne, Switzerland), PermaNet^®^ 3.0 (containing deltamethrin coated on the net’s polyester side panels and a mixture of deltamethrin and PBO on the polyethylene roof) (Vestergaard, Lausanne, Switzerland), Olyset^®^Net (2% permethrin) (Sumitomo Chemical UK PLC, London, UK) and Olyset^®^ plus (containing 2% permethrin combined with 1% of PBO in the whole net) (Sumitomo Chemical UK PLC, London, UK). At the same time, mosquitoes exposed to untreated nets were used as controls. After exposure, mosquitoes were removed, kept in a paper cup and provided with sugar solution and the mortality rate was scored after 24 h.

### 2.5. Expression Profile of D7 Salivary Genes Using Real-Time Quantitative PCR

The expression profiles of members of the D7 family in insecticide resistant and susceptible mosquitoes of *An. funestus s.s.* and *An. gambiae* species were assessed using real-time quantitative polymerase chain reaction (qPCR). Taking into account the existence of two D7 subfamilies (long and short forms genes families) in mosquitoes [3], eight genes were used for *An. gambiae*: three long forms (*D7L1-3*) and five short forms (*D7r1-5*), while five were targeted for *An. funestus s.s*: one long form (*D7L*) and four short forms (*D7r1-4*). Two pairs of exon-spanning primers was designed for each gene using Primer3 online software (v4.0.0; http://bioinfo.ut.ee/primer3/). Only primers with PCR efficiency between 90 and 110% determined using a cDNA dilution series obtained from a single sample, were used for qPCR analysis. Details of the primers used for this study are listed in Appendix A.

To run the analysis, total RNA was extracted for each species from three batches of 10 whole females 3–5 days old from the following sample sets: (i) for resistant (R) strain, alive mosquitoes after exposure to 0.75% permethrin and 0.05% deltamethrin; (ii) for the control (C) strain, unexposed mosquitoes to insecticides and thus representative of the wild-type population; and (iii) unexposed mosquitoes from the fully susceptible laboratory strains (S), FANG (for *An. funestus*), KISUMU (for *An. gambiae*). RNA was isolated using the RNAeasy Mini kit (Qiagen, Hiden, Germany) according to the manufacturer’s instructions. The RNA quantity was assessed using a NanoDrop ND1000 spectrophotometer (Thermo Fisher, Waltham, MA, USA) and 1 µg from each of the three biological replicates for resistant (R), control (C) and susceptible (S) for both species was used as a template for cDNA synthesis using the SuperScript III (Invitrogen, Carlsbad, CA, USA) with oligo-dT20 and RNase H, following the manufacturer’s instructions. The qPCR assays were carried out in a MX 3005 real-time PCR system (Agilent, Santa Clara, CA, USA) using Brilliant III Ultra-Fast SYBR Green qPCR Master Mix (Agilent). A total of 10 ng of cDNA from each sample was used as template in a three-step program involving a denaturation at 95 °C for 3 min followed by 40 cycles of 10 s at 95 °C and 10 s at 60 °C and a last step of 1 min at 95 °C, 30 s at 55 °C and 30 s at 95 °C. The relative expression and fold-change of each target gene in R and C relative to S was calculated according to the 2^−ΔΔCT^ method incorporating PCR efficiency after normalization with the housekeeping RSP7 ribosomal protein S7 (VectorBase ID: AFUN007153; orthologous in *An. gambiae*: AGAP010592) and the actin 5C (vectoBase ID: AFUN006819, orthologous in *An. gambiae*: AGAP000651) genes for *An. funestus* and ii) RSP7 ribosomal protein S7 (vectoBase ID:AGAP010592) and Glyceraldehyde 3-phosphate dehydrogenase, GADPH (vectoBase ID:AGAP009945) for *An. gambiae*.

### 2.6. Sequencing of D7r3 and D7r4 Genomic DNA from Alive and Dead *An. funestus* Mosquitoes

DNA extracted using the LIVAK method from a total of 20 *An. funestus s.s* female specimens (10 alive and 10 dead after exposure to deltamethrin) was used as template for amplification and sequencing of a region of 974 bp (from 528 to 1501) and 951 bp (from 432 to 1382) from the genomic DNA sequences of *D7r3* and *D7r4* genes, respectively. The primers used were designed with the Primer3 online software (v4.0.0; http://bioinfo.ut.ee/primer3/) and the sequences are listed in Appendix A. DNA amplification, sequencing and analysis were carried out following a protocol previously described [23].

### 2.7. Genotyping of Resistance Molecular Markers in An. funestus s.s. and in An. gambiae

The L1014F-kdr and the L1014S-kdr mutations responsible for Dichlorodiphenyltrichloroethane (DDT) and pyrethroid resistance in *An. gambiae* were genotyped in field collected mosquitoes using the protocol described by Martínez-Torres and colleagues [24]. TaqMan assays were performed for genotyping of A296S-RDL marker involved in the resistance to dieldrin in *An. funestus s.s* following a protocol previously described [25]. On the other hand, an allele specific PCR was performed as previously described [26,27] to genotype the L119F-GSTe2 mutation which is associated to DDT/pyrethroid resistance in *An. funestus* vector species. For these markers, the genotyping was carried out using genomic DNA extracted from field-collected mosquitoes.

## 3. Results

### 3.1. Mosquito Species Composition

A total of 328 blood-fed female mosquitoes were collected indoor in Elon. 256 (78%) were morphologically identified as belonging to the *An. funestus* group, whereas 64 (19.5%) belonged to the *An. gambiae* complex and the remaining 8 (2.5%) were *Culex* spp. Of the 173 *An. funestus s.l* randomly selected and tested for molecular identification, *An. funestus s.s.* represented 95.95% and the remaining was constituted by *An. rivulorum* (1.73%), *An. parensis* (1.16%) and *An. rivulorum-like* (1.16%). Concerning *An. gambiae sl*, all the 64 individuals collected were identified as *An. gambiae*.

### 3.2. Plasmodium Infection

For both *An. funestus* and *An. gambiae*, head-thorax from field-collected mosquitoes were separately used to assess the *Plasmodium* infection rate. For *An. funestus*, the analysis of head and thorax (153 individuals) revealed 21 mosquitoes infected (13.7%), with 20 (95%) infected with *P. falciparum* and one with either *P. ovale*, *P. malariae*, *P. vivax*. Regarding *An. gambiae*, seven (20.6%) out of the 34 head and thorax analyzed were infected only by *P. falciparum*.

### 3.3. Insecticide Susceptibility Bioassays

#### 3.3.1. Insecticide Susceptibility in *An. funestus s.s.*

A total of 765 F_1_ female mosquitoes were exposed to seven insecticides (Figure 1A). *An. funestus* mosquito from Elon were resistant to all types of pyrethroids tested. A mortality rate of 62.5 ± 9.2% and 36.25 ± 6.9% was recorded after exposure to permethrin and deltamethrin, respectively. For etofenprox a pseudo-pyrethroid, mortality rate of 56.4 ± 13.4% was observed. Regarding organochlorine, a full resistance was recorded for dieldrin. The exposure to carbamates indicated a susceptibility to propoxur (98.95 ± 1.25%), whereas a possible resistance was noticed for bendiocarb (91.7 ± 4.4% of mortality). A full susceptibility was observed after exposure to fenitrothion and to malathion (100% mortality).

#### 3.3.2. Insecticide Susceptibility in *An. gambiae*

For *An. gambiae*, a total of 200 F_1_ female mosquitoes were exposed to permethrin and deltamethrin only (Figure 1B). For both insecticides, a high resistance was observed, as only 3.33 ± 3.3% and 1.67 ± 1. 67% mortality rates were recorded for permethrin and deltamethrin, respectively.

#### 3.3.3. Synergist Assays for Pyrethroid Resistance in *An. funestus s.s.*

To assess whether cytochrome P450s enzymes are involved in the resistance to pyrethroids in *An. funestus s.s* from Elon, a total of 160 F_1_ female mosquitoes were exposed to deltamethrin and permethrin following pre-exposure to PBO. A high recovery of susceptibility was observed for both insecticides with 100% for deltamethrin and 97.5 ± 1.44% for permethrin (Figure 1A).

#### 3.3.4. Insecticide Treated bed nets Efficacy Assessment

A loss of efficacy against *An. funestus* was observed for the two brands of bed nets mostly used in Cameroon, with only 33.75 ± 3.75% and 27.50 ± 2.50% mortality after 3 min exposure to Olyset and PermaNet 2.0, respectively. A high but not full recovery of the efficacy was observed with Olyset Plus (71.25 ± 1.25% mortality) and PermaNet 3.0 side (57.50 ± 13.15% mortality), while a full recovery was noticed for PermaNet 3.0 roof with 100% mortality (Figure 1C).

For *An. gambiae*, because of low density of mosquitoes, only PermaNet brands bed nets were tested. A loss of efficacy was observed for both PermaNet 2.0 (5 ± 2.9% mortality) and PermaNet 3.0 side (20 ± 10% mortality), whereas PermaNet 3.0 still led to 100% mortality of field mosquitoes after exposure (Figure 1D).

### 3.4. Expression on D7 Salivary Genes Family in An. funestus and *An. gambiae* Mosquitoes

Analysis of the expression level of *D7* genes in *An. funestus* from Elon indicated that *D7r2*, *D7r3* and *D7r4* are over-expressed in both permethrin and deltamethrin resistant field *An. funestus* mosquitoes compared to susceptible strain (Figure 2). However, this over-expression was statistically significant only for both *D7r3* and *D7r4* genes but not for *D7r2* (Table 1 and Table 2).

Indeed, a fold-change of 11.99 (*p* = 0.04) and 4.44 (*p* = 0.006) for *D7r3* and 6.24 (NS) and 4.012 (*p* = 0.02) for *D7r4*, were observed when comparing samples from Elon with FANG strain respectively for permethrin resistant and non-exposed mosquitoes (Figure 2A and Table 1). A similar pattern of over-expression of *D7r3* and *D7r4* was observed in deltamethrin resistant (FC = 9.88, *p* = 0.001 for *D7r3*; FC = 4.45, *p* = 0.026 for *D7r4*) and unexposed (FC = 8.4, *p* < 0.001 for D7r3; FC = 6.76, *p* = 0.04 for *D7r4*) mosquitoes compared to the susceptible strain (Figure 2B and Table 2).

For *An. gambiae*, only three genes (*D7r1*, *D7r2* and *D7r3*) were expressed in both permethrin resistant (field samples) and susceptible (KISUMU) strains (Figure 3A), whereas, for deltamethrin, *D7r1*, *D7r2*, *D7r3* and *D7r4* genes were expressed in resistant and susceptible mosquitoes (Figure 3b). Globally, members of D7 family genes were under-expressed in permethrin/deltamethrin resistant and unexposed mosquitoes from Elon compared to the susceptible strain (Figure 3A,B). This under-expression was significant for *D7r2* gene in permethrin (FC = 0.26, *p* = 0.007 for resistant and FC = 0.27, *p* = 0.008 for non-exposed mosquitoes) and deltamethrin (FC = 0.16, *p* = 0.001 and FC = 0.25; *p* = 0.02 for resistant and non-exposed) field resistant mosquitoes (Table 1 and Table 2). *D7r3* (FC = 0.54, *p* = 0.0002 and FC = 0.44; *p* = 0.006 for resistant and non-exposed respectively) and *D7r4* genes (FC = 0.26, *p* = 0.01 and FC = 0.34; *p* = 0.006 for resistant and non-exposed respectively) were significantly under-expressed only in deltamethrin resistant and non-exposed field mosquitoes compared to susceptible strain (Table 1 and Table 2).

### 3.5. Sequencing of *D7r3* and *D7r4* Genomic DNA from *An. funestus* Mosquitoes

A comparative analysis of the polymorphism pattern of both *D7r3* and *D7r4* genes, significantly over-expressed in resistant mosquitoes, was carried out between sets of 10 alive and 10 dead mosquitoes after exposure to deltamethrin. A 718bp common sequence of *D7r3* gene was aligned for 13 individuals (7 alive and 6 dead), while for *D7r4* the alignment was done on a 776 bp common sequence for 11 individuals (4 alive and 7 dead). Overall, 84 and 77 polymorphic sites defining 18 haplotypes were detected for *D7r3* and *D7r4*, respectively. The number of haplotypes (9) for *D7r3* gene was similar between dead and alive mosquitoes, whereas for D7r4, alive mosquitoes showed a lower number of polymorphic sites with a reduced haplotype diversity (4 haplotypes; Hd = 0.86) compared to dead individuals (14 haplotypes; Hd = 1). The high haplotype diversity was associated with high genetic diversity for both genes (Table 3 and Table 4).

Analysis of the haplotypes network for both *D7r3* and *D7r4* showed no major haplotypes of these genes in *An. funestus* population from Elon (Figure 4a,b). Overall, Tajima’s D index (D = −1.04 for *D7r3* and D = −0.98 for *D7r4*) were negative but not statistically significant for both genes whereas, the Fu’s F index (F = −1.54 for *D7r3* and F = −2.63 for *D7r4*) even negative for both genes, was significant only for *D7r4*. The maximum likelihood phylogenetic tree generated for both genes did not show any haplotype clustering associated with either dead or alive mosquitoes (Figure 4c,d).

### 3.6. Molecular Basis of the Insecticide Resistance in Field Malaria Vector Populations

#### 3.6.1. RDL-A296S and L119F-Gste2 mutations detection in *An. funestus s.s.*

From a total of 190 field collected *An. funestus* mosquito randomly genotyped for RDL-A296S mutation, 160 (87%) were homozygous resistant (RR) and 24 (13%) were homozygous susceptible (SS). Concerning the L119F-GSTe2 mutation, 11/73 (15%), 35/73 (48%) and 27/73 (37%) individuals were carried the homozygous resistant (RR), the heterozygote (RS) and the susceptible (SS) genotype, respectively.

#### 3.6.2. L1014F mutations in *An. Gambiae*

The *kdr* mutation was genotyped in a total of 60 mosquitoes. Of these, 56 (93.3%) were homozygous resistant (RR) and 4 (6.7%) were heterozygous (RS).

## 4. Discussion

Assessing mechanisms which are associated with the spreading of insecticide resistance in *Anopheles* mosquitoes is crucial in understanding its impact on malaria transmission. It was reported that insecticide resistance in mosquitoes may impact on their feeding habit and vector competence [28]. Mosquito blood meal intake and pathogens transmission are facilitated by various salivary proteins through their anti-coagulant, anti-platelet aggregation and immunosuppressive properties that help the mosquito to overcome homeostasis and blood feeding [3,4,29]. However, few information are available on the impact that insecticide resistance may have on the salivary proteins of malaria vectors. With the purpose to help filling this gap of knowledge, the present study investigated the potential impact of insecticide resistance on the expression and genetic diversity of *D7* gene family.

Because no study had previously been conducted in the study area, we firstly characterized the malaria vector populations in the locality by determining their composition specificity as well as their susceptibility to the main insecticides used in public health. The morphological identification of collected mosquitoes revealed that *Anopheles funestus s.l.* and *Anopheles gambiae* complex were the sole *Anopheline* species found indoor of the habitations. *Anopheles funestus* appeared to be the most abundant malaria vector in Elon, accounting for 78% of the collected mosquitoes. This pattern of malaria vectors composition and abundance is similar to previous reports from a study in Nkoteng, an area located 50 km from Elon [30] but this could vary according to climatic season. Moreover, because the collection method we used specifically targeted indoor resting mosquitoes, the present study is limited on giving an accurate overview of the *Anopheline* species composition in the study area. The species-typing PCR assays led to the identification of *An. funestus s.s* as the predominant member of *An. funestus* group in the study area. This observation is in accordance with what described in several previous studies carried out in Cameroon [27,30,31,32]. Nonetheless, even if detected in in very low proportion, the presence of other species such as, *An. parensis, rivulorum* and *rivulorum-like*, indicates some diversity in the species composition of *An. funestus* group in the study area. This species diversity highlights the need of accurate identification of this species group to generate reliable data on these malaria vectors. For the *An. gambiae* complex, *An. gambiae* was the only species found. This is probably linked to the fact that the study is carried in a rural area where *An. gambiae* is more adapted than *An. coluzzii* [33].

With the exception of organophosphates and propoxur (carbamates), the *An. funestus* mosquito population from Elon exhibited resistance towards several insecticides tested such as deltamethrin, permethrin, dieldrin and bendiocarb. A high level of resistance to deltamethrin and permethrin was also observed for *An. gambiae s.s.* This multiple resistance profile is in line with what was previously reported in different studies in the country [27,31,34,35,36]. This confirms that resistance to several insecticides is widespread in malaria vectors populations in Cameroon. Results of the cone bioassays emphasize with both *An. funestus s.s.* and *An. gambiae s.s.* species, showed a low mortality rates with both permethrin (Olyset) and deltamethrin (PermaNet) impregnated LLINs. This reduction of bio-efficacy has been previously hypothesized to be due to the selection pressure induced by the massive distribution of bed nets by the government associated to use of pesticide in agriculture [27]. This reduction of bio-efficacy could favor malaria transmission in this area where high Plasmodium infection rates were observed in the main vectors. Nevertheless, distribution of nets with PBO could represent an alternative in this area since almost full recovery of susceptibility to both deltamethrin and permethrin was observed in *An. funestus* after a pre-exposure to PBO confirming the implication of cytochrome P450s in insecticide resistance of *An. funestus* populations from Elon as previously observed in other locations in the country [27,31]. However, the high frequency of *kdr* mutation detected in these mosquito populations indicates that pyrethroid resistance in these malaria vectors is due to both metabolic resistant and *kdr* mutation in Elon. This observation confirms the results of several other authors [27,31,34,37,38,39].

The comparative expression profiling of the *D7* family genes in both species in relation to susceptible strains suggests a potential influence of insecticide resistance on some of these genes in *An. funestus*. The qPCR analyses showed that the expression of *D7* family genes is associated with resistance to the two pyrethroids mostly used in malaria vector control tools in Cameroon. Such association was previously detected by microarray analyses in different studies across the African continent [5,6,7,8,9,10]. Furthermore, a recent study using both microarray and qPCR analysis have reported an overexpression of *D7* family genes in bendiocarb resistant *An. gambiae* mosquito from Uganda [40]. In this previous study, the authors hypothesized that *D7* proteins may confer insecticide resistance by binding and sequestering insecticide or insecticide metabolites rather than by any direct detoxification. They supported their hypothesis by implementing in silico experimentations to demonstrate the structural compatibility of the *D7r4* protein’s central binding pocket and bendiocarb. Additionally, they suggested that the larger pocket found in *D7r2* may permit binding of pyrethroids as reported for cytochrome P450 enzymes such as *CYP6M2* [41,42]. However, at this stage, no experimental evidence has been generated to prove the involvement of *D7* genes in detoxification of insecticide. In our study, a significant overexpression was observed for two short form genes (*D7r3* and *D7r4*) in both permethrin and deltamethrin resistant *An. funestus* field mosquitoes. The overexpression of *D7r3* gene was previously reported after microarray assay in bendiocarb and DDT resistant *An. funestus* mosquito populations from Malawi [43] and Benin [6]. On the other hand, to our knowledge, this is the first time a differential expression of *D7r4* gene has been reported in the insecticide resistant *An. funestus* mosquito. Further investigations are needed to confirm these results. The *D7r2* gene was also found overexpressed in both permethrin and deltamethrin resistant *An. funestus* mosquito. This overexpression, even if non-significant in the present study, is in accordance with previous microarray studies in permethrin and lambda-cyhalotrin resistant *An. funestus* population in Malawi [43] in Zambia [44], respectively. These results seem to indicate an association of insecticide resistance with an overexpression of some *D7* family genes in *An. funestus* mosquito from multiple regions of Africa. However, in this stage of the study, we cannot hypothesize whether or not these genes are implicated in the development of pyrethroids resistance in *An. funestus* or if it is a result of the mosquito’s physiological adaptation to resistance.

To investigate whether the overexpression of *D7r3* and *D7r4* in *An. funestus* is related to a polymorphism of those genes, we sequenced a full length DNA of both genes from alive and dead mosquitoes after exposure to deltamethrin. Although high haplotype diversity was observed for both genes, no major haplotype was found associated with either resistant (alive) or susceptible (dead) phenotypes. This indicates that for both genes, there is no selection of a specific haplotype associated to insecticide resistance and consequently, to their over-expression. The high haplotype diversity coupled to the high genetic diversity observed for both genes would suggest that this *An. funestus* population is expanding as a result of a recent event within these genes. This hypothesis is supported principally for the *D7r4* gene, by the negative and statistically significant value of the Fu’s F index. Further studies would be more informative on how important is the expansion of this *An. funestus* population from Elon.

Regarding *Anopheles gambiae*, we observed an under-expression of overall members of *D7* family genes in both the permethrin and deltamethrin resistant field strain compared to the Kisumu lab strain. These results are contrasting with the overexpression observed from microarray analyses in pyrethroids resistant *An. gambiae* mosquito from Zambia [44], in *An. coluzzii* from Burkina-Faso and Côte-d’Ivoire [8] and in *An. arabiensis* from Sudan [45] or in Uganda [9] and Zanzibar [5]. The difference between these previous results and what we observed in the present study could be due to specificity of the primers used for qPCR for this species. So, there is a need of further studies to confirm the pattern of differential expression of *D7* family genes in pyrethroids resistant *An. gambiae* populations potentially with new primer sets. However, the fact that, both microarray and qPCR assays recently revealed an overexpression of *D7r2* and *D7r4* genes in bendiocarb resistant *An. gambiae* populations from Uganda [40] suggests that *D7* is also associated with resistance in this species. Although the difference between these results and ours could be linked to insecticide tested. Indeed, because mosquitoes develop different resistance mechanisms to carbamates and pyrethroids, the impact of the resistance to one family could be different to what observed for the other. However, the down-regulation of *D7* family genes observed in our study is in accordance with results of previous studies after comparative proteomic analysis of expressed proteins in bendiocarb susceptible and resistant mosquitoes [46,47]. Indeed, in these studies conducted in *Culex quinquefasciatus* [48] and *An. gambiae* [47], the D7 long form (the sole member of *D7* family protein detected) presented lower expression in the resistant strain than the susceptible strain. Globally, it appears that the pattern of how insecticide resistance is associated to the differential expression of *D7* family genes remained unclear. This highlights the need to conduct both transcriptomic and proteomic further studies to draw a definite conclusion.

The present study provides some important information that could contribute to fill the gap on knowledge on the impact of insecticide resistance in salivary gland proteins. Nevertheless, certain aspects not taken into account here would contribute to improve the information provided by the present study. Indeed, qPCR analyses were not carried out in all the members of *D7* genes for *An. funestus*, as we did not succeed to have primers for *D7r5* and *D7L2* for this species. Moreover, it would certainly be more informative to work with resistant and susceptible mosquitoes coming from the same locality, instead of comparing field resistant mosquitoes with lab colonized susceptible strain with very low genes polymorphism as done in the present study.

## 5. Conclusions

In the present study, after assessing insecticide susceptibility in malaria vectors in Elon, we compared the expression of *D7* family genes in the salivary gland of pyrethroid resistant and susceptible *An. gambiae* and *An. funestus s.s.* mosquitoes. We found that *An. gambiae* and *An. funestus s.s.* are both multiple resistant to almost all the insecticides currently used in public health. A significant over-expression of two short forms of the *D7* family genes (*D7r3* and *D7r4*) in *An. funestus s.s.*, while an under-expression was observed for almost all the D7 genes in *An. gambiae* This result confirms that insecticide resistance could be associated to the expression of the salivary proteins in malaria vectors. Therefore, because salivary proteins are essential for mosquito blood meal intake and for malaria parasites transmission, additional studies are needed to assess whether the differential expression of D7 genes due to insecticide resistance statute could have an impact on the blood intake success and in the ability of *Anopheles* mosquitoes to be infected by malaria sporozoites. Such studies will be informative on the profile of malaria transmission in the context of insecticide resistance, as is the case in many countries in Africa. Furthermore, studies exploring how insecticide resistance is associated with the differential expression of salivary glands genes may open the way for screening of novel insecticide resistance candidate genes, as current known resistance mutations explain only a fraction of a resistance phenotype.

## Figures and Tables

**Figure 1 genes-10-00211-f001:**
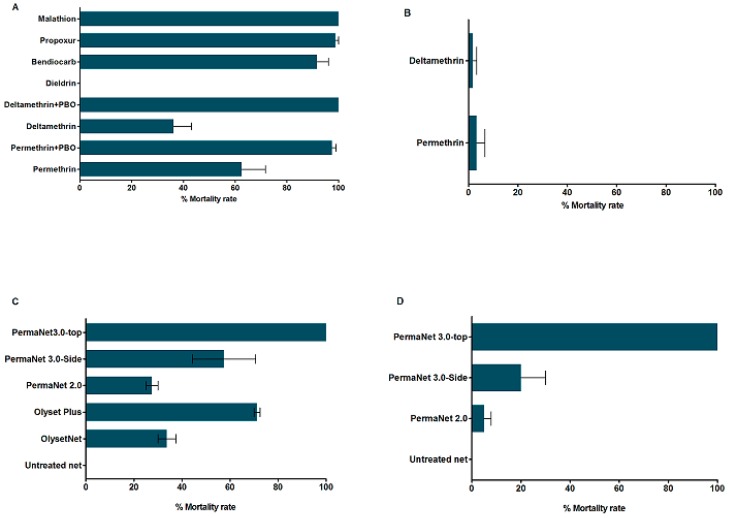
Susceptibility status of the *An. funestus* and *An. gambiae* populations from Elon and bio-efficacy of mosquito bed nets: Mortality rates of *An. funestus s.s.* (**A**) and of *An. gambiae* (**B**) with World Health Organization (WHO) tube assays after 1 h exposure to insecticides. Recorded mortalities rates following 3-min exposure by cone assays with various nets for *An. funestus* (**C**) and *An. gambiae* (**D**). Data are shown as mean ± SEM.

**Figure 2 genes-10-00211-f002:**
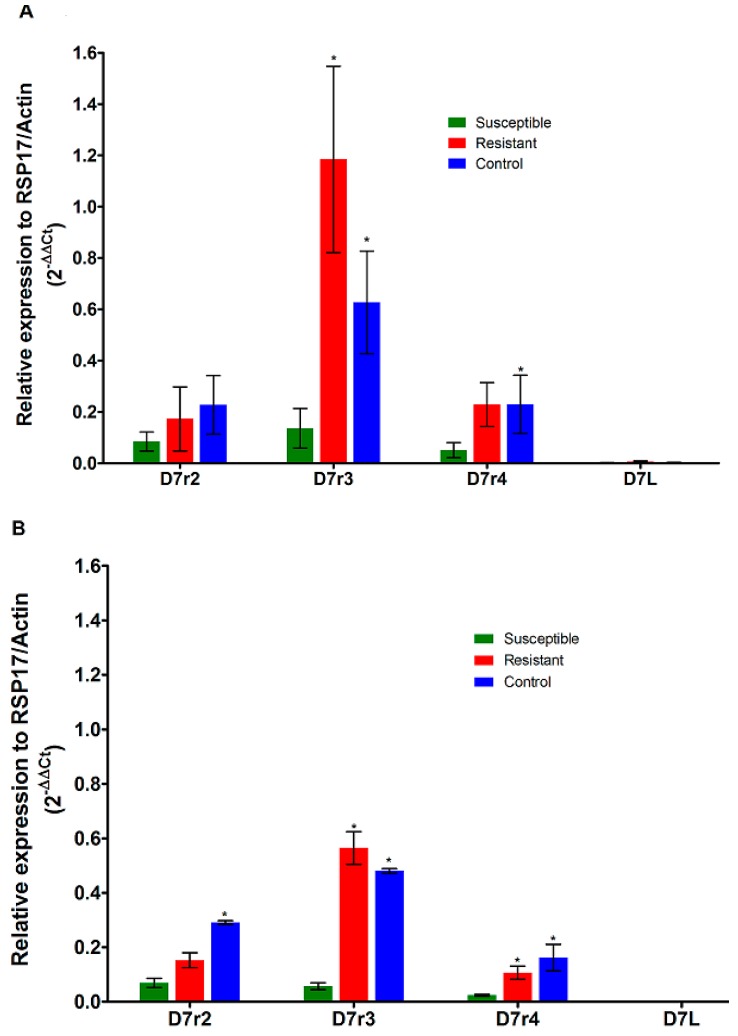
Transcription profile of members of *D7* family genes in *An. funestus s.s* population. Comparison of the patterns of genes expression of four *D7* genes (*D7r2*, *D7r3*, *D7r4* and *D7L*) between field permethrin (**A**) or deltamethrin (**B**) resistant mosquitoes, field mosquitoes unexposed to insecticides (Control) and the laboratory susceptible strain FANG (Susceptible). The normalized relative expression of each gene against two housekeeping genes (*RSP7* and Actin) is represented on the vertical axis. (*) indicates a significant differential expression in comparison with susceptible mosquito.

**Figure 3 genes-10-00211-f003:**
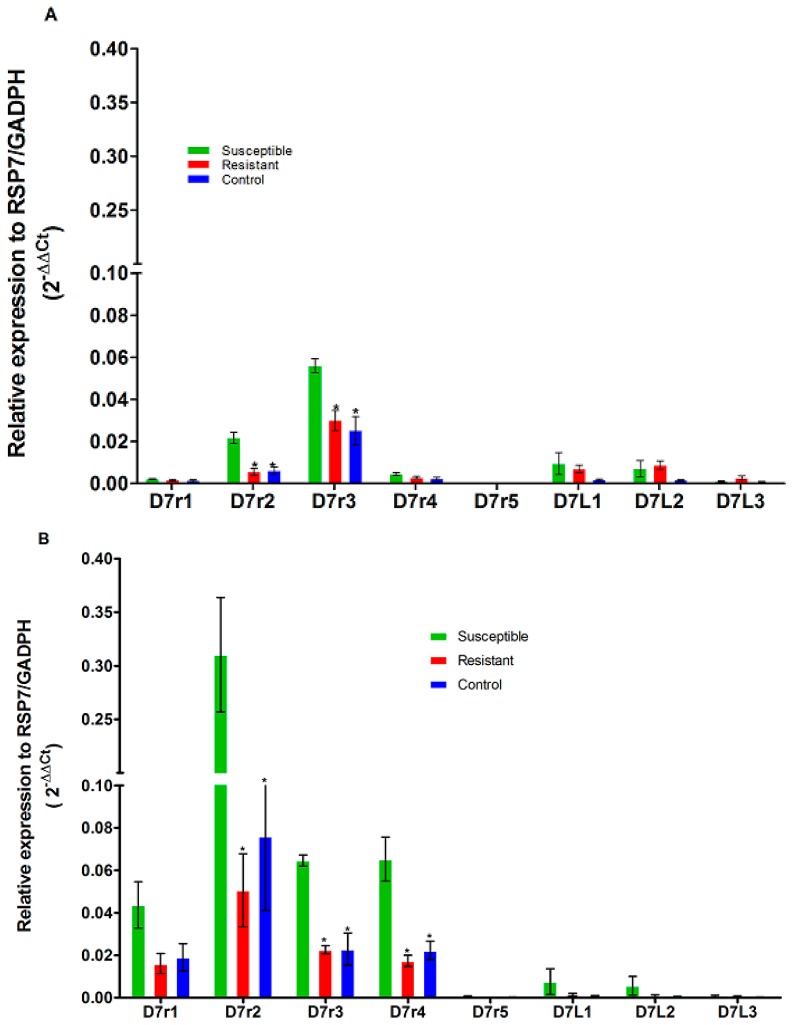
Transcription profile of members of D7 family genes in *An. gambiae* population. Comparison of the patterns of genes expression of eight D7 genes (*D7r2*, *D7r3*, *D7r4*, *D7r5*, *D7L1*, *D7L2* and *D7L3*) between field permethrin (**A**) or deltamethrin (**B**) resistant mosquitoes, field mosquitoes unexposed to insecticides (Control) and the laboratory susceptible strain FANG (Susceptible). The normalized relative expression of each gene against two housekeeping genes (*RSP7* and *GADPH*) is represented on the vertical axis. (*) indicates a significant differential expression in comparison with susceptible mosquito.

**Figure 4 genes-10-00211-f004:**
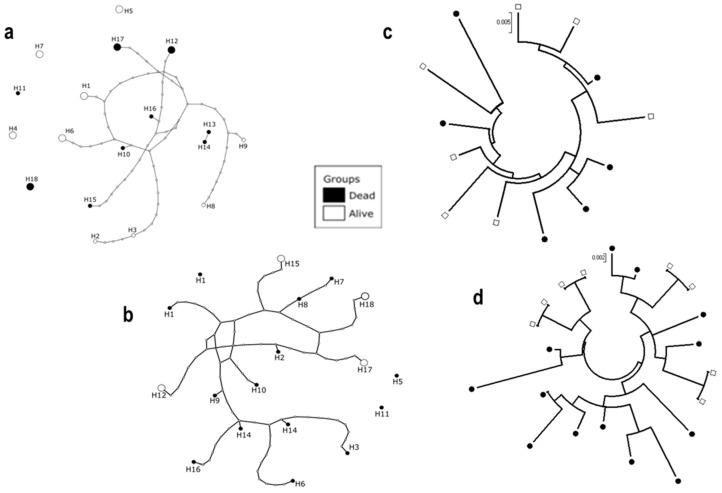
Analysis of the polymorphism of the *D7r3* and *D7r4* genes in *An. funestus s.s.* mosquito. Haplotype network showing high diversity in *D7r3* (**a**) and *D7r4* (**b**) genes. Maximum likelihood phylogenetic tree of *D7r3* (**c**) and *D7r4* (**d**) genes showing a lack of association between haplotypes and resistance phenotype to deltamethrin.

**Table 1 genes-10-00211-t001:** Differential expression of some *D7* family genes in permethrin resistant and unexposed *An. funestus s.s.* and *An. gambiaes s.s.* mosquitoes, as measured by qPCR analyses.

Species	Gene	Fold Change Resistant vs. Susceptible	Log2FC	*p*-Value	Fold Change Control vs. Susceptible	Log2FC	*p*-Value
*An. funestus*	*D7r2*	6.66	2.73	NS	2.24	1.16	NS
	*D7r3*	11.99	3.58	0.04	4.44	2.15	0.006
	*D7r4*	6.24	2.64	NS	4.012	2	0.02
*An. gambiae*	*D7r1*	0.744	−0.425	NS	0.659	−0.60	NS
	*D7r2*	0.257	−1.95	0.007	0.27	−1.84	0.008
	*D7r3*	0.535	−0.90	0.01	0.44	−1.15	0.01

**Table 2 genes-10-00211-t002:** Differential expression of some *D7* family genes in deltamethrin resistant and unexposed *An. funestus s.s.* and *An. gambiae*. mosquitoes, as measured by qPCR analyses.

Species	Gene	Fold Change Resistant vs. Susceptible	Log2FC	*p*-Value	Fold Change Control vs. Susceptible	Log2FC	*p*-Value
*An. funestus*	*D7r2*	2.2	1.1	NS	4.19	2.06	0.0002
	*D7r3*	9.88	3.304	0.001	8.404	3.07	<0.0001
	*D7r4*	4.45	2.15	0.026	6.76	2.75	0.04
*An. gambiae*	*D7r1*	0.36	−1.44	NS	0.434	−1.20	NS
	*D7r2*	0.163	−2.61	0.001	0.245	−2.02	0.02
	*D7r3*	0.349	−1.51	0.0002	0.353	−1.5	0.006
	*D7r4*	0.26	−1.91	0.01	0.34	−1.55	0.01

**Table 3 genes-10-00211-t003:** Summary statistics for polymorphism in *D7r3* gene in deltamethrin susceptible and resistant *An. funestus s.s.* from Elon.

	2N	H	S	Hd	π	D	D *
Alive	14	9	52	0.956	0.00142	0.175	1.132
Dead	12	9	49	0.955	0.0022	−0.53006	0.0895
Total	26	18	84	0.978	0.024	−1.04	−1.54

2N, number of sequences; D, Tajima’s statistics; D *, Fu and Li’s statistics; H, number of haplotypes; Hd, haplotype diversity; π, nucleotide diversity multiplied by 103; S, number of polymorphic sites.

**Table 4 genes-10-00211-t004:** Summary statistics for polymorphism in *D7r4* gene in deltamethrin susceptible and resistant *An. funestus s.s.* from Elon.

	2N	H	S	Hd	π	D	D *
Alive	8	4	28	0.857	0.017	0.98	5.27 *
Dead	14	14	67	1	0.022	−0.95	−4.19 *
Total	22	18	77	0.983	0.022	−0.98	−2.610 *

2N, number of sequences; D, Tajima’s statistics; D *, Fu and Li’s statistics; H, number of haplotypes; Hd, haplotype diversity; π, nucleotide diversity multiplied by 103; S, number of polymorphic sites.

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
