# Peer review of "Overexpression of Two Members of D7 Salivary Genes Family is Associated with Pyrethroid Resistance in the Malaria Vector Anopheles Funestus s.s. but Not in Anopheles Gambiae in Cameroon"

_genes, 2019, doi:10.3390/genes10030211_

Round 1

Reviewer 1 Report

The manuscript entitled: “Overexpression of two members of D7 salivary genes family is associated with pyrethroid resistance in the malaria vector Anopheles funestus s.s. but not in Anopheles gambiae in Cameroon” by Emmanuel Elanga-Ndille and colleagues describes the expression changes of D7 family members in the salivary gland of pyrethroid resistant and susceptible An. gambiae and An. funestus s.s. mosquitoes, after assessing insecticide susceptibility in Elon, Cameroon.

While in An. gambiae a recent comprehensive microarray analysis (Isaac and colleague, 2018, also cited by authors) described transcriptional modulation of salivary gland genes from mosquitoes displaying resistance to bendiocarb in Uganda, much less information is available for An. funestus.

As a general comment, in my opinion this work provides the scientific community with results that offer only some degree of novelty considering the existing literature. Indeed, the main conclusion that members of the D7 family are overexpressed in resistant Anopheles mosquitoes is already presented in a few publications, as mentioned by authors in the Discussion. On the other hand, the results obtained by authors concerning An. gambiae are in contrast to existing literature and this discrepancy is essentially attributed by authors to technical issues associated to the primer sequences. There are not novel evidences trying to shed light on the mechanisms underlying the expression changes (role of promoters? Enhancers? Epigenetic factors?) nor on the possible role an increased amount of D7 transcripts might have in the insecticide resistance (a direct, “scavenging” role as hypothesized in Isaac et al., 2018, or a behavioral change in mosquito blood-feeding habit). 

However, the preliminary accurate description of insecticide resistance/susceptibility in Elon, Central Cameroon linked to the precise expression analysis approach, i.e. Real Time PCR, make the story of interest for field scientists and suitable for publications, after a few revisions will be accomplished.

1)     In order to understand whether insecticide resistance alters blood-feeding behavior by modulating expression of salivary genes, a deeper approach (as in Isaacs et al, 2018) would have been clearly preferable, but an improvement of the current approach would be anyway favorable. For instance: i) the selection of a higher number of salivary glands-specific genes among those with a function assigned (not only members of the D7 family), or ii) at least, and I think this is important for the acceptance of the manuscript, the analysis of the complete D7 family. Indeed, authors stated: “In An. funestus, members of D7 gene family are currently known to be constituted by four genes coding for short proteins (D7r1, D7r2, D7r3 and D7r4) and one coding for long protein (D7L) [5, 12-14]”. However, the recent sequencing of 16 anophelinae genomes and annotation of salivary genes (Neafsey et al., Science 2015 and Arcà et al., BMC Genomics 2017) allowed to update the list of D7 proteins in An. funestus, with three long and five short forms identified as in An. gambiae. According to VectorBase, AFUN016458 is the ortholog of D7r5, and AFUN007417 is the ortholog of D7L2, while the other D7 long form it was found only as a transcript. To have a complete picture of the transcriptional modulation of a gene cluster such as the D7 cluster, I believe it would be important to analyze the expression profile of all the members of the cluster (7 genes in around 40 kbp), since they show a very similar tissue-specific expression pattern (restricted to female salivary glands) making possible the hypothesis of a common mechanism of expression regulation.

2)     Please, detail better the procedure for Real Time quantification. How exactly (using which formula?) are the Ct values normalized with two endogenous reference genes? I guess that “incorporating PCR efficiency” stated in the Methods means that PCR efficiency of the different primer pairs were sufficiently similar and that the error was anyway calculated as a function of the PCR efficiencies: is this correct? Moreover, DDCT means that DCT values after normalization are also calibrated (DDCT), so that one selected sample chosen as calibrator takes the value of “1” when exponentiation is performed. Is it a calibrator included in your analysis (not shown in the graphs)? Finally, please organize better the graphs in Figure 3, possibly by cutting the y-axes, otherwise the differences among columns are not visible.

Minor revisions and typos.

Rows 35-36: please, check punctuation.

Rows 66-68: please, check the sentence.

Rows 75-76: please, check the sentence.

Row 175: please, add AGAP ID codes for An. funestus.

Rows 359-361: please, check this sentence.

Author Response

1)     In order to understand whether insecticide resistance alters blood-feeding behavior by modulating expression of salivary genes, a deeper approach (as in Isaacs et al, 2018) would have been clearly preferable, but an improvement of the current approach would be anyway favorable. For instance: i) the selection of a higher number of salivary glands-specific genes among those with a function assigned (not only members of the D7 family),

Answer: The reviewer proposed that one good approach to understand whether insecticide resistance alters Anopheles blood-finding behavior by modulating expression of salivary genes, could consist on the selection of a higher number of salivary glands-specific genes among those with a function assigned. We totally agree with him as, D7 proteins family are not the sole proteins described to be implicated in the blood-feeding process of mosquitoes. However, we chose in the present study to focus only on D7 family proteins for the following reasons:

- In the literature on the transcriptomic analysis comparing insecticide resistant and susceptible malaria vectors, the D7 family genes are the salivary gland genes with assigned function that are mostly reported to be differently expressed.

- In a recent study of Isaacs and colleagues, among the salivary gland genes detected, the D7 family genes were the ones found associated to insecticide resistance using both microarray and qPCR. However, in this study, the authors were focused only on An. gambiae sl mosquitoes resistant to carbamate. Because mechanisms involved on the resistance are not the same for all insecticide classes, we aimed in the present study to assess if the expression of D7 family genes could also be modulated by pyrethroids which are massively used on bednets and for which malaria vectors had developed a resistance mechanism different from the one used against carbamates.

- D7 family proteins are among the most abundantly expressed genes in the salivary gland of mosquitoes such as Anopheles and which are reported to be involve in blood feeding process

or ii) at least, and I think this is important for the acceptance of the manuscript, the analysis of the complete D7 family. Indeed, authors stated: “In An. funestus, members of D7 gene family are currently known to be constituted by four genes coding for short proteins (D7r1, D7r2, D7r3 and D7r4) and one coding for long protein (D7L) [5, 12-14]”. However, the recent sequencing of 16 anophelinae genomes and annotation of salivary genes (Neafsey et al., Science 2015 and Arcà et al., BMC Genomics 2017) allowed to update the list of D7 proteins in An. funestus, with three long and five short forms identified as in An. gambiae. According to VectorBase, AFUN016458 is the ortholog of D7r5, and AFUN007417 is the ortholog of D7L2, while the other D7 long form it was found only as a transcript. To have a complete picture of the transcriptional modulation of a gene cluster such as the D7 cluster, I believe it would be important to analyze the expression profile of all the members of the cluster (7 genes in around 40 kbp), since they show a very similar tissue-specific expression pattern (restricted to female salivary glands) making possible the hypothesis of a common mechanism of expression regulation.

Answer: We agree with the reviewer about the new annotation findings. We did miss these other members of D7 family genes in An. funestus. As, the reviewer said, it would indeed have been interesting to analyze the expression of all the members of the cluster to have a complete picture of the transcriptional modulation of D7 family genes like we did for An. gambiae. Unfortunately, because we do not have the samples (cDNA) used anymore, we are not able to run new qPCR analyses with these two genes. Indeed these samples were used for other analyses such as the quantification of the expression of some detoxification enzymes associated to insecticide resistance.  However, aware that this is a limit for our study, we have not only rewritten lines 79-81, but also highlight it in lines 446-453 in the discussion section.

2)     Please, detail better the procedure for Real Time quantification. How exactly (using which formula?) are the Ct values normalized with two endogenous reference genes?

Answer: The analyses of our qPCR data were performed according the following steps:

- Calculation of the Ct values according to PCR efficiency correction. Formula used:

CtE = Ct*((log(E+1)/(log(2)) 

- Average the technical replicates

- Calculation on the Delta Ct value (Ct Target - Ct Control). This was done for each of the two reference genes used. 

- Calculation of the delta delta Ct value (2^-dCt) for each reference gene separately

- Calculation of the delta delta Ct value (2^-dCt COMBINED) for both reference genes: (DDCt-Reference gene 1+ DDCt-Reference gene 2)/ 2

-  Calculation of the average of delta delta Ct (2-dCt Mean) for Biological replicates combining both reference genes

- Calculation of fold changes. This was done by dividing the average DDCt of each sample by the average DDCt of the sample used as calibrator (susceptible mosquito).

I guess that “incorporating PCR efficiency” stated in the Methods means that PCR efficiency of the different primer pairs were sufficiently similar and that the error was anyway calculated as a function of the PCR efficiencies: is this correct?  

Answer: Here the efficiency was determined from results of analysis of the standard curve as given by the Mx3005 qPCR machine. Only one efficiency was given for a pair of primers (forward and reverse). Only primers with PCR efficiency between 90 and 110% were used in the present study. Thereafter, raw Ct values were corrected with the PCR efficiency of each gene using the following formula: CtE = Ct*((log(E+1)/(log(2))

Moreover, DDCT means that DCT values after normalization are also calibrated (DDCT), so that one selected sample chosen as calibrator takes the value of “1” when exponentiation is performed. Is it a calibrator included in your analysis (not shown in the graphs)?  

Answer: Yes we used a calibrator for our analysis. The selected sample chosen as calibrator was the “susceptible” mosquito. When calculating the fold changes, it took the value 1 as its average DDCt was divided by itself.

Finally, please organize better the graphs in Figure 3, possibly by cutting the y-axes, otherwise the differences among columns are not visible.

Answer: This was taking into account and the graphs have been reorganized as recommended by the reviewer

Minor revisions and typos.

Rows 35-36: please, check punctuation.

Answer: The sentence was rewritten as followed: “All An. gambiae s.l mosquitoes were identified as An. gambiae. High levels of pyrethroid resistance were observed in both An. gambiae and An. funestus mosquitoes

Rows 66-68: please, check the sentence.

Answer: The sentence was rewritten as followed:  “Interestingly, several studies assessing the transcriptomic changes associated with insecticide resistance have detected beside the overexpression of detoxification genes, a differential expression of some genes encoding for salivary proteins such as, D7 salivary proteins [6-11]

Rows 75-76: please, check the sentence.

Answer: The sentence was modified as followed: “The D7 salivary proteins family in Anopheles mosquito species was described to comprise several genes. These genes”

Row 175: please, add AGAP ID codes for An. funestus.

Answer: This comment was taking into account and the ID codes were added for An. funestus see lines 175-178 in the revised manuscript.

Rows 359-361: please, check this sentence.

Answer: The sentence was modified as followed: “Nonetheless, even if detected  in very low proportion, the detection of the presence of other species such as, An. parensis, rivulorum and rivulorum-like, indicates some diversity in the species composition of An. funestus group in the study area”.

Reviewer 2 Report

The authors present a very interesting research comparing the expression of the D7 salivary protein genes in insecticide resistant and susceptible vectors of malaria in Africa. Their findings suggest an association between resistant mosquitoes and expression changes of genes coding D7 short proteins. The study was carefully designed, and proper controls were used to determine the significance of the findings. They caught mosquitoes from the field and use the first generation to determine resistance, then evaluate D7 gene expression.

Abstract
Lines 35 to 36 needs revision. Misplaced periods.

Lines 38 – 41. A paragraph describing the significance of these findings in missing in the abstract. What are these results suggesting, what is the potential association between the resistance and blood meal intake?

Methods
Lines 116-119. Was this essay done in pools or testing individual mosquitoes? This should be clarified in this section.

Results

Authors evaluated the presence of Plasmodium parasites in the collected mosquitoes, but the results of this testing are not shown. It would be very helpful to see the gene expression of the D7 genes in mosquitoes infected compared to the presence/absence of resistance.

Discussion

 Although the authors did an excellent and through job discussing their results, a small paragraph summarizing the limitation of the study will be helpful to put the findings into the proper context.

Author Response

The authors present a very interesting research comparing the expression of the D7 salivary protein genes in insecticide resistant and susceptible vectors of malaria in Africa. Their findings suggest an association between resistant mosquitoes and expression changes of genes coding D7 short proteins. The study was carefully designed, and proper controls were used to determine the significance of the findings. They caught mosquitoes from the field and use the first generation to determine resistance, then evaluate D7 gene expression.

Abstract
Lines 35 to 36 needs revision. Misplaced periods.

Answer: This observation was taking into account and the sentence was rewritten as followed:The sentence rewritten as followed: All An. gambiae s.l mosquitoes were identified as An. gambiae. High levels of pyrethroid resistance were observed in both An. gambiae and An. funestus mosquitoes

Lines 38 – 41. A paragraph describing the significance of these findings in missing in the abstract. What are these results suggesting, what is the potential association between the resistance and blood meal intake?

Answer: We thank the reviewer for this comment. However we would like to indicate that in lines 38-41 of the submitted manuscript, we have highlighted the potential involvement of the differential expression of D7 gene due to insecticide resistance on the blood meal intake. Because we didn’t implemented complementary experiment to assess the impact on the blood meal intake, we wanted to avoid talking more about it. However we said something on this aspect as followed: “However, no association was observed between the polymorphism of these genes and their over-expression. In contrast, overall D7 salivary genes were under-expressed in pyrethroid resistant An. gambiae. This study provides preliminary evidences that pyrethroid resistance could influence blood meal intake through over-expression of D7 proteins although future studies will help establishing potential impact on vectorial capacity.  

Methods
Lines 116-119. Was this essay done in pools or testing individual mosquitoes? This should be clarified in this section.

Answer: To be more precise, the sentence was modified in the revised manuscript as followed:Plasmodium infection rate of both An. funestus group and An. gambiae complex mosquitoes was determined using a TaqMan assay to detect four Plasmodium species (Plasmodium falciparum, Plasmodium vivax, Plasmodium ovale, and Plasmodium malariae) [21] from genomic DNA extracted from individual field-collected mosquitoes”. See lines: 117-120

Results

Authors evaluated the presence of Plasmodium parasites in the collected mosquitoes, but the results of this testing are not shown.

Answer: The result of Plasmodium infection is presented in the section 3.2. Plasmodium infection in lines 227-233 of the submitted manuscript.

It would be very helpful to see the gene expression of the D7 genes in mosquitoes infected compared to the presence/absence of resistance.

Answer: We agree with the reviewer for this comment, but this was not feasible in our case because the Plasmodium infection in the present study was assessed using field collected mosquitoes, whereas expression of D7 genes was done in F1 mosquitoes obtained in the lab. However, we are currently running Plasmodium experimental infection using resistant and susceptible mosquitoes selected from new field collected mosquitoes. These mosquitoes will be used to assess the association between resistance and Plasmodium infection.

Discussion

 Although the authors did an excellent and through job discussing their results, a small paragraph summarizing the limitation of the study will be helpful to put the findings into the proper context

Answer: This comment was taking into account and a short paragraph was included in the discussion section. Lines 446-453

Reviewer 3 Report

This manuscript by Elanga-Ndille and others describes teh possible correlation of D7 salivary genes in insecticide resistant in An. funestus and An. gambiae.

A very descriptive work that would need, in my opinion, more substantial data to conclusively demonstrate the role of salivary D7 proteins in insecticide resistance. The hypothesis that D7 salivary proteins could be in any way in detoxification of insecticides is based on a "in silico" analysis and hasn't experimentally been proved. I strongly suggest the authors to reword this claim in the Discussion section noting that this has not been proven.

The manuscript makes reference to the D7 short forms in Anopheline mosquitoes, however, the authors failed to cite 2 articles published by Dr. Ribeiro (PMID: 16301315) and Dr. Andersen groups (PMID: 17928288), specially when discussing the possibility of D7r2/4 binding insecticide molecules (line 381). I strongly recommend the authors to cite these 2 papers.

In the supplementary file (list of primers used for qPCR) the authors listed the wrong accession number for D7r3. The correct accession number is AGAP008283. Because of this error, I checked the D7r3 in Vector Base and the forward primer was found but i was unable to find the reverse primer. It might be a typo but the authors need to correct this mistake.

Also, I recommend the authors to list the amplicon size for the qPCR products. The recommended amplicon size for accurate qPCR reactions should be no longer than 80-100mer.

Author Response

This manuscript by Elanga-Ndille and others describes teh possible correlation of D7 salivary genes in insecticide resistant in An. funestus and An. gambiae.

A very descriptive work that would need, in my opinion, more substantial data to conclusively demonstrate the role of salivary D7 proteins in insecticide resistance. The hypothesis that D7 salivary proteins could be in any way in detoxification of insecticides is based on a "in silico" analysis and hasn't experimentally been proved. I strongly suggest the authors to reword this claim in the Discussion section noting that this has not been proven.

Answer: We thank the reviewer for this comment and we agree with him about the hypothesis that the D7 proteins could be involved in detoxification of insecticide is still based on a silico analysis. In our study, we did no declared that this hypothesis is proven, we just mentioned it in relation to the suggestion made by Issacs et al. It can be seen in our text that this remains only an hypothesis stated by others  (Isaacs et al). We have not validated this hypothesis as this was not the purpose our study.  However, to avoid any confusion, this part of the manuscript was modified in the revised manuscript as followed: “In previous study, authors hypothesized that D7 proteins may confer insecticide resistance by binding and sequestering insecticide or insecticide metabolites rather than by any direct detoxification. They supported their hypothesis by implementing in silico experimentations to demonstrate the structural compatibility of the D7r4 protein’s central binding pocket and bendiocarb. Additionally, they suggested that the larger pocket found in D7r2 may permit binding of pyrethroids as reported for cytochrome P450 enzymes such as CYP6M2 [42, 43]. However, at this stage, no experimental evidence has been given to prove the involvement of D7 genes in detoxification of insecticide”. See lines : 391 - 398

The manuscript makes reference to the D7 short forms in Anopheline mosquitoes, however, the authors failed to cite 2 articles published by Dr. Ribeiro (PMID: 16301315) and Dr. Andersen groups (PMID: 17928288), specially when discussing the possibility of D7r2/4 binding insecticide molecules (line 381). I strongly recommend the authors to cite these 2 papers.

Answer: The PMID 16301315 indicated by the reviewer corresponds to a paper published by Dr Ribeiro team in 2006. The first author of this paper is Calvo E and the paper was cited in our submitted manuscript as reference n°4. However, while reading the comment of the reviewer, we thought this paper must also be referenced when talking about D7 genes in line 78 of the manuscript. This is now done.

Regarding the second paper (PMID: 17928288, Mans et al, 2007) it is now cited in the revised manuscript.  See lines 78-79

In the supplementary file (list of primers used for qPCR) the authors listed the wrong accession number for D7r3. The correct accession number is AGAP008283. Because of this error, I checked the D7r3 in Vector Base and the forward primer was found but i was unable to find the reverse primer. It might be a typo but the authors need to correct this mistake.

Answer: We would like to thank the reviewer for this remark. Indeed, it’s an error when typing the number. The correct accession number (AGAP008283) is now indicated in the supplementary file.

Also, I recommend the authors to list the amplicon size for the qPCR products. The recommended amplicon size for accurate qPCR reactions should be no longer than 80-100mer.

Answer: This recommendation of the reviewer was taking into account and a column showing amplicon size has been added in the list of primers used for qPCR. However, according to the book written by Tania Nolan and collaborators in 2013 (T Nolan, J Huggett, E Sanchez, Good practice guide for the application of quantitative PCR (qPCR), LGC (2013)), we choose to work with amplicon size comprised between 80-150 bases.

Round 2

Reviewer 1 Report

In the revised version of the manuscript entitled: “Overexpression of two members of D7 salivary genes family is associated with pyrethroid resistance in the malaria vector Anopheles funestus s.s. but not in Anopheles gambiae in Cameroon” authors replied to the criticisms raised in the first revision step.

They recognize a limit of the study since the lack of cDNAs makes not possible to perform new experiments. However, the data presented for the D7 members analyzed are interesting and worth publication.

Regarding my concerns about Real Time quantification, authors state in the response letter:

Answer: Yes we used a calibrator for our analysis. The selected sample chosen as calibrator was the “susceptible” mosquito. When calculating the fold changes, it took the value 1 as its average DDCt was divided by itself.

Maybe I’m not getting properly this issue, but I still don’t see a single green column (related to susceptible samples) in Figure 2 and 3 set to value 1. Could you please explain why?